# On (α,β)-US Sets in *BCK/BCI*-Algebras

**Chiranjibe Jana *** and **Madhumangal Pal**

Department of Applied Mathematics with Oceanology and Computer Programming, Vidyasagar University, Midnapore 721102, India; mmpalvu@gmail.com
* Correspondence: jana.chiranjibe7@gmail.com; Tel.: +91-9647135650

**Abstract:** Molodtsov originated soft set theory, which followed a general mathematical framework for handling uncertainties, in which we encounter the data by affixing the parameterized factor during the information analysis. The aim of this paper is to establish a bridge to connect a soft set and the union operations on sets, then applying it to $BCK/BCI$-algebras. Firstly, we introduce the notion of the $(\alpha, \beta)$-Union-Soft ($(\alpha, \beta)$-US) set, with some supporting examples. Then, we discuss the soft $BCK/BCI$-algebras, which are called $(\alpha, \beta)$-US algebras, $(\alpha, \beta)$-US ideals, $(\alpha, \beta)$-US closed ideals, and $(\alpha, \beta)$-US commutative ideals. In particular, some related properties and relationships of the above algebraic structures are investigated. We also provide the condition of an $(\alpha, \beta)$-US ideal to be an $(\alpha, \beta)$-US closed ideal. Some conditions for a Union-Soft (US) ideal to be a US commutative ideal are given by means of $(\alpha, \beta)$-unions. Moreover, several characterization theorems of (closed) US ideals and US commutative ideals are given in terms of $(\alpha, \beta)$-unions. Finally, the extension property for an $(\alpha, \beta)$-US commutative ideal is established.

**Keywords:** $BCK/BCI$-algebra; $(\alpha, \beta)$-US set; $(\alpha, \beta)$-US subalgebra; $(\alpha, \beta)$-US (closed) ideal; $(\alpha, \beta)$-US commutative ideal

**MSC:** 06F35; 03G25; 06D72

## 1. Introduction

Most of the real-world problems in social sciences, the environment, engineering, medical sciences, economics, etc., involve data that contain uncertainties. To overcome these uncertainties, researchers are motivated to introduce some classical theories like the theories of fuzzy sets [1], rough sets [2], intuitionistic fuzzy sets [3], vague sets [4], and interval mathematics [5], i.e., by which we can have a mathematical tool to deal with uncertainties. Even though sets are very powerful to model the problems containing uncertainties, in some cases, these sets are not enough to overcome the serious types of uncertainties experienced in real-world problems. In that situation, in 1999, Molodtsov [6] posited the novel concept of soft set theory, which is a completely new approach for modeling vagueness and uncertainties. Some tremendous developments based on soft sets [7] have been recently drawn the attention of many scholars: in particular, Aktas and Çagman [8] used soft groups; Acar et al. [9] provided soft rings; and Ali et al. [10] defined new working rules on soft sets. Çagman et al. [11] studied soft-intgroups. Feng et al. [12] described soft semi-rings. Sezgin et al. [13] described soft intersection near-rings. Now, the above results have been applied to the disciplines of information sciences, decision support systems, knowledge systems, decision-making, and so on, reviewed in [14–20]. Recently, researchers have drawn attention to modeling covering-based problems on rough and soft sets with their applications to MADMproblems [21–23]. Additionally, there is a huge scope of application to develop covering-based algebraic structures in different kinds of algebras. Zhan et al. [24–26] introduced a new notion called $(M, N)$-SI-h-bi-ideals and

$(M, N)$-SI-h-quasi-ideals in the environment of hemirings. At the same time, different soft algebraic structures have been developed on $BCK/BCI$-algebras, which was proposed by Imai and Iśeki [27,28]. Here, we briefly review some results of soft sets in the existing literature of $BCK/BCI$-algebras. Jana et al. [29–31], Ma and Zhan [32–34], and Senapati et al. [35,36] performed detailed investigations on $BCK/BCI$-algebras and related algebraic systems. In [37], Jun first constructed soft algebraic structure of $BCK/BCI$-algebras. Jun and Park [38] also pointed out applications of soft sets in the ideal theory of $BCK/BCI$-algebras. Jun et al. [39] also studied the soft $p$-ideal of soft $BCI$-algebras. Acar and Özürk [40] analytically studied maximal, irreducible, and prime soft ideals of $BCK/BCI$-algebras with supporting examples. First, Jun et al. [41,42] proposed a novel concept, namely union-soft sets and int-soft sets, and then implemented it to develop union-soft $BCK/BCI$-algebras and int-soft $BCK/BCI$-algebras. Sezgin [43] considered studying soft union interior ideals, quasi-ideals, and generalized bi-ideals of rings and gave their interrelationship. She also studied regular, intra-regular, regular-duo, and strongly-regular properties of rings in terms of soft-union ideals. Sezgin et al. [44] introduced a new soft classical ring theory, namely soft intersection rings, ideals, bi-ideals, interior ideals, and quasi-ideals. Furthermore, they defined their soft-union intersection product and their corresponding relationships. Jana and Pal [45] defined the concept of $(\alpha, \beta)$-soft intersection sets and then introduced this ideal to develop $(\alpha, \beta)$-soft intersectional groups structures and their various properties. Jana and Pal [46] also motivated using the same concept for the development of $(\alpha, \beta)$-soft intersectional $BCK/BCI$ algebraic structures. Again, Jana et al. [47] proposed providing $(\alpha, \beta)$-soft intersectional rings, $(\alpha, \beta)$-soft intersectional ideals, and their relationships. In this environment, different union-soft algebras and $(\alpha, \beta)$-soft intersectional algebras in different uncertain fuzzy environments under soft operations are considered by us as enough motivation to develop our proposal. There is an important issue in defining new $(\alpha, \beta)$-US sets based on soft operations and their application to develop different kinds of US algebraic structures. Therefore, based on the $(\alpha, \beta)$-US operation, how to develop $(\alpha, \beta)$-US $BCK/BCI$-algebras is a tremendous topic. To solve this problems, in this paper, we shall develop $(\alpha, \beta)$-US $BCK/BCI$-algebras introducing the concept of $(\alpha, \beta)$-US sets and their application to subalgebras, ideals, and commutative ideals in $BCK/BCI$-algebras on the basis of traditional union-soft algebras [24,25,41,42] and the results of [26,48]. We discuss the relationship between $(\alpha, \beta)$-US subalgebras, $(\alpha, \beta)$-US ideals, and $(\alpha, \beta)$-US commutative ideals in detail. We provide the condition that an $(\alpha, \beta)$-US ideal is an $(\alpha, \beta)$-US commutative ideal.

The remainder of this article is structured as follows: Section 2 proceeds with a recapitulation of all required definitions of $BCK/BCI$-algebras, basic definitions of soft sets, and subsequent discussions. In Section 3, the concepts of $(\alpha, \beta)$-US sets are proposed and illustrated by some examples. In Section 4, the notion of $(\alpha, \beta)$-US subalgebras of $BCK/BCI$-algebras is introduced and their properties discussed in detail. In Section 5, some interesting properties of $(\alpha, \beta)$-US ideals in $BCK/BCI$-algebras are introduced. Some characterization theorems of the $(\alpha, \beta)$-US commutative ideals in $BCK/BCI$-algebras are established in Section 6. Finally, in Section 7, conclusions and the scope for future research are given.

## 2. *BCK/BCI*-Algebras and Soft Sets

In this section, we introduce some elementary aspects that are necessary for this paper. For more information regarding $BCK/BCI$-algebras, the reader is referred to the monograph [49]. By a $BCI$-algebra, we mean an algebra $(X, *, 0)$ of the type $(2, 0)$ satisfying the following axioms for all $x, y, z \in X$:

$(C_1)$ $((x * y) * (x * z)) * (z * y) = 0$
$(C_2)$ $(x * (x * y)) * y = 0$
$(C_3)$ $x * x = 0$
$(C_4)$ $x * y = 0$ and $y * x = 0$ imply $x = y$.

If a *BCI*-algebra $X$ satisfies the following identity:

$(C_5)$  $0 * x = 0,$

then $X$ is called a *BCK*-algebra. Any *BCK/BCI*-algebra satisfies the following axioms: for all $x, y, z \in X$

$(C_6)$  $x * 0 = x$
$(C_7)$  $x \leq y \Rightarrow x * z \leq y * z, z * y \leq z * x$
$(C_8)$  $(x * y) * z = (x * z) * y$
$(C_9)$  $(x * z) * (y * z) \leq x * y$

The partial ordering is defined as $x \leq y$ if and only if $x * y = 0$. In a *BCI*-algebra $X$, the following hold:

$(C_{10})$  $(x * (x * (x * y))) = x * y$
$(C_{11})$  $(0 * (x * y)) = (0 * x) * (0 * y).$

A non-empty subset $S$ of a *BCK/BCI*-algebra $X$ is called a subalgebra of $X$ if $x * y \in S$ for all $x, y \in X$. A *BCK*-algebra $X$ is said to be commutative $x \wedge y = y \wedge x$ for all $x, y \in X$, where $y \wedge x = y * (y * x)$. A subset $I$ of a *BCK/BCI*-algebra $X$ is called an ideal of $X$ if for all $x, y \in X$,

$(C_{12})$  $0 \in A$
$(C_{13})$  $y \in A$ and $x * y \in A \Rightarrow x \in A.$

A subset $I$ of a *BCK*-algebra $X$ is called a commutative ideal if it satisfies $(C_{12})$ and for $z \in I$
$(C_{14})$ $((x * y) * z \in I \Rightarrow (x * (y * (y * x))) \in I.$
An ideal $I$ of a *BCK*-algebra $X$ is called commutative if it satisfies the implication $x * y \in I \Rightarrow x * (y * (y * x)) \in I.$

$U$ refers to an initial universal set, and $E$ is the set of parameters. Let $\mathcal{P}(U)$ be the power set of $U$ and $A \subset E$. Molodtsov [6] introduced soft sets in the following manner:

**Definition 1** ([6]). *A pair $(\mathcal{F}, E)$ is called a soft set over $U$ if $\mathcal{F}$ is a function given by:*

$$\mathcal{F} : E \to \mathcal{P}(U).$$

*In other words, a soft set in the universe $U$ is a parameterized family of subsets of the universal set $U$. For $\varepsilon \in A$, $\mathcal{F}(\varepsilon)$ may be considered as the set of $\varepsilon$-elements of the soft $(\mathcal{F}, A)$ or as the set of $\varepsilon$-approximate elements of the soft set.*

The following example illustrates the above idea.

**Example 1.** *Let $(X, \tau)$ be a topological space, i.e., $\tau$ is a family of subsets of the set $X$ called the open sets of $X$. Then, the family of open neighborhoods $N(x)$ of point $x$, where $N(x) = \{V \in \tau | x \in V\}$, may be considered as the soft set $(N(x), \tau)$.*

**Definition 2** ([6,14]). *For a non-empty subset $A$ of $E$, a soft set $(\mathcal{F}, E)$ over $U$ satisfying the condition:*

$$\mathcal{F}(x) = \varnothing \; for \; all \; \; x \notin A$$

*is called the A-soft set over U and is denoted by $\mathcal{F}_A$, so an A-soft set $\mathcal{F}_A$ over U is a function $\mathcal{F}_A : E \to \mathcal{P}(U)$ such that $\mathcal{F}_A(x) = \varnothing$ for all $x \notin A$. A soft set over U can be followed by the set of ordered pairs:*

$$\mathcal{F}_A = \{(x, \mathcal{F}_A(x)) : x \in E, \mathcal{F}_A(x) \in \mathcal{P}(U)\}.$$

We remark that a soft set is a parameterized family of subsets of the set U. A soft set $\mathcal{F}_A(x)$ may be an arbitrary, empty, and nonempty intersection. The set of all soft sets over U is denoted by $S(U)$.

**Definition 3** ([14]). *Let $\mathcal{F}_A \in S(U)$. For all $x \in E$, If $\mathcal{F}_A(x) = \varnothing$, then $\mathcal{F}_A$ is said to be an empty soft set and symbolized by $\Phi_A$. If $\mathcal{F}_A(x) = U$, then $\mathcal{F}_A$ is said to be an A-universal soft set and symbolized as $\mathcal{F}_{\widetilde{A}}$. If $\mathcal{F}_A(x) = U$ and $A = E$, then $\mathcal{F}_{\widetilde{A}}$ is said to be a universal soft set and is denoted by $\mathcal{F}_{\widetilde{E}}$.*

**Proposition 1** ([14]). *Let $\mathcal{F}_A \in S(U)$. Then,*
*(i) $\mathcal{F}_A \widetilde{\cup} \mathcal{F}_A = \mathcal{F}_A$, $\mathcal{F}_A \widetilde{\cap} \mathcal{F}_A = \mathcal{F}_A$.*
*(ii) $\mathcal{F}_A \widetilde{\cup} \Phi_A = \mathcal{F}_A$, $\mathcal{F}_A \widetilde{\cap} \Phi_A = \Phi_A$.*
*(iii) $\mathcal{F}_A \widetilde{\cup} \mathcal{F}_E = \mathcal{F}_E$, $\mathcal{F}_A \widetilde{\cap} \mathcal{F}_E = \mathcal{F}_A$.*
*(iv) $\mathcal{F}_A \widetilde{\cup} \mathcal{F}_A^c = \mathcal{F}_E$, $\mathcal{F}_{\widetilde{A}}^c \widetilde{\cup} \mathcal{F}_A^c = \Phi_A$, where $\Phi_A$ is an empty set.*

**Definition 4** ([37]). *Let E be a BCK/BCI-algebra and $(\mathcal{F}, A)$ be a soft set over BCK/BCI-algebra E. Then, $(\mathcal{F}, A)$ is called a soft BCK/BCI-algebra over E if $\mathcal{F}(x)$ is a subalgebra of E for all $x \in E$.*

**Definition 5** ([42]). *Let E be a BCK/BCI-algebra. Let $\mathcal{F}_A \in S(U)$ for a given subalgebra A of E. Then, $\mathcal{F}_A$ is called a US algebra of A over U if, for all $x, y \in A$, it satisfies the following condition:*

$$\mathcal{F}_A(x * y) \subseteq \mathcal{F}_A(x) \cup \mathcal{F}_A(y).$$

**Definition 6** ([42]). *Let E be a BCK/BCI-algebra and A be a subalgebra of E. Let $\mathcal{F}_A \in S(U)$. Then, $\mathcal{F}_A$ is called a US ideal over U if, for all $x, y \in A$, it satisfies the following conditions:*
*(1) $\mathcal{F}_A(0) \subseteq \mathcal{F}_A(x)$*
*(2) $\mathcal{F}_A(x) \subseteq \mathcal{F}_A(x * y) \cup \mathcal{F}_A(y)$.*

**Definition 7** ([42]). *Let E be a BCK/BCI-algebra. For a given subalgebra A of E, let $\mathcal{F}_A \in S(U)$, then the US ideal $\mathcal{F}_A$ is said to be closed if, for all $x \in A$, it satisfies the following condition:*

$$\mathcal{F}_A(0 * x) \subseteq \mathcal{F}_A(x).$$

**Definition 8** ([42]). *Let E be a BCK-algebra. For a given subalgebras A of E, let $\mathcal{F}_A \in S(U)$. Then, $\mathcal{F}_A$ is called a US commutative ideal over U if, for all $x, y, z \in A$, it satisfies the following conditions:*
*(1) $\mathcal{F}_A(0) \subseteq \mathcal{F}_A(x)$*
*(2) $\mathcal{F}_A(x * (y * (y * x))) \subseteq \mathcal{F}_A((x * y) * z) \cup \mathcal{F}_A(z)$.*

**Definition 9** ([42]). *Let $\mathcal{F}_A \in S(U)$ and $\delta \subseteq U$. Then, the $\delta$-exclusion set of $\mathcal{F}_A$, denoted by $\mathcal{F}_A^\delta$, is defined by $\mathcal{F}_A^\delta(x) = \{x \in A | \mathcal{F}_A(x) \subseteq \delta\}$.*

## 3. $(\alpha, \beta)$-US Sets

In this section, $U$ is the initial universe, $E$ is the set of parameters, and "$\rightarrowtail$" is a binary operation. Now, we let $S(U)$ be the set of all soft sets. We introduce the notion of $(\alpha, \beta)$-US sets and illustrate them by some examples. From now on, we let $\varnothing \subseteq \alpha \subset \beta \subseteq U$.

**Definition 10.** *For any non-empty subset $A$ of $E$, consider the soft set $\mathcal{F}_A \in S(U)$. Then, for all $x, y \in A$, the soft set $\mathcal{F}_A$ is called an $(\alpha, \beta)$-US set over $U$ if it satisfies the following condition:*

$$\mathcal{F}_A(x \rightarrowtail y) \cap \beta \subseteq \mathcal{F}_A(x) \cup \mathcal{F}_A(y) \cup \alpha.$$

**Example 2.** *We consider five houses in the initial universe set $U$, which is given by*

$$U = \{h_1, h_2, h_3, h_4, h_5\}.$$

*Let the set of parameters $E = \{\xi_1, \xi_2, \xi_3, \xi_4\}$ be the status of the set of houses, which follows for the parameters "cheap", "expensive:, "in the flooded area:, and "in the urban area", respectively, with the following binary operation:*

| $\rightarrowtail$ | $\xi_1$ | $\xi_2$ | $\xi_3$ | $\xi_4$ |
|---|---|---|---|---|
| $\xi_1$ | $\xi_1$ | $\xi_1$ | $\xi_1$ | $\xi_1$ |
| $\xi_2$ | $\xi_2$ | $\xi_1$ | $\xi_1$ | $\xi_2$ |
| $\xi_3$ | $\xi_3$ | $\xi_3$ | $\xi_1$ | $\xi_3$ |
| $\xi_4$ | $\xi_4$ | $\xi_4$ | $\xi_4$ | $\xi_1$ |

*We consider a soft set $\mathcal{F}_E$ over $U$, which is given as $\mathcal{F}_E(\xi_1) = \{h_3, h_5\}$, $\mathcal{F}_E(\xi_2) = \{h_3, h_4, h_5\}$, $\mathcal{F}_E(\xi_3) = \{h_2, h_3, h_4, h_5\}$, and $\mathcal{F}_E(\xi_4) = \{h_1, h_3, h_5\}$. Fix $\beta = \{h_1, h_2, h_3, h_5\}$ and $\alpha = \{h_2, h_3\}$. Then, it can be easily verified that $\mathcal{F}_E$ is an $(\alpha, \beta)$-US set over $U$.*

**Theorem 1.** *Let $\mathcal{F}_A, \mathcal{F}_B \in S(U)$ be soft sets such that $\mathcal{F}_A$ is a soft subset of $\mathcal{F}_B$. If $\mathcal{F}_B$ is an $(\alpha, \beta)$-US set over $U$, then the same holds for $\mathcal{F}_A$.*

**Proof.** Let $x, y \in A$ such that $x \rightarrowtail y \in A$. Then, $x \rightarrowtail y \in B$ since $A \subseteq B$. Thus, $\mathcal{F}_A(x \rightarrowtail y) \cap \beta \subseteq \mathcal{F}_B(x \rightarrowtail y) \cap \beta \subseteq \mathcal{F}_B(x) \cup \mathcal{F}_B(y) \cup \alpha = \mathcal{F}_A(x) \cup \mathcal{F}_A(y) \cup \alpha$. Therefore, $\mathcal{F}_A$ is an $(\alpha, \beta)$-US set over $U$.  □

The converse of Theorem 1 is not true in general, as can be seen in the following example.

**Example 3.** *We consider five houses in the initial universe set $U$, which is given by:*

$$U = \{h_1, h_2, h_3, h_4, h_5\}.$$

*Let the set of parameters $E = \{\xi_1, \xi_2, \xi_3, \xi_4\}$ be the status of the set of houses, which follows for the parameters "beautiful", "cheap", "in a good location", and "in green surroundings", respectively, with the following binary operation:*

| $\rightarrowtail$ | $\xi_1$ | $\xi_2$ | $\xi_3$ | $\xi_4$ |
|---|---|---|---|---|
| $\xi_1$ | $\xi_1$ | $\xi_2$ | $\xi_3$ | $\xi_4$ |
| $\xi_2$ | $\xi_2$ | $\xi_1$ | $\xi_3$ | $\xi_2$ |
| $\xi_3$ | $\xi_3$ | $\xi_4$ | $\xi_1$ | $\xi_2$ |
| $\xi_4$ | $\xi_4$ | $\xi_3$ | $\xi_2$ | $\xi_1$ |

*Let $A = \{\xi_1, \xi_2\} \subset E$. Consider a soft set $\mathcal{F}_E$ over $U$ as $\mathcal{F}_A(\xi_1) = \{h_1, h_3\}$, $\mathcal{F}_A(\xi_2) = \{h_1, h_3, h_4\}$, $\mathcal{F}_A(\xi_3) = \varnothing$, $\mathcal{F}_A(\xi_4) = \varnothing$, $\beta = \{h_1, h_3, h_4, h_5\}$, and $\alpha = \{h_3, h_4\}$. Then, it can be easily verified that $\mathcal{F}_A$ is an $(\alpha, \beta)$-US set over $U$.*

*Consider another soft set $\mathcal{F}_B$ as $\mathcal{F}_B(\xi_1) = \{h_1, h_3\}$, $\mathcal{F}_B(\xi_2) = \{h_1, h_3, h_4\}$, $\mathcal{F}_B(\xi_3) = \{h_2, h_4\}$, and $\mathcal{F}_B(\xi_4) = \{h_4, h_5\}$. Then, $\mathcal{F}_A$ is a soft subset of $\mathcal{F}_B$. However, for $\beta = \{h_1, h_3, h_4, h_5\}$ and $\alpha = \{h_3, h_4\}$, $\mathcal{F}_B$ is not an $(\alpha, \beta)$-union soft set over $U$, because:*

$$\mathcal{F}_B(\xi_3 \rightarrowtail \xi_4) \cap \beta = \{h_1, h_3, h_4\} \nsubseteq \{h_2, h_3, h_4, h_5\} = \mathcal{F}_B(\xi_3) \cup \mathcal{F}_B(\xi_4) \cup \alpha.$$

### 4. $(\alpha, \beta)$-US Subalgebras in *BCK/BCI*-Algebras

In this section, we introduce the concept of the $(\alpha, \beta)$-US subalgebra of *BCK/BCI*-algebras and investigate some of its characterization. Throughout this section, $E = X$ is always a *BCK/BCI*-algebra without any specification.

**Definition 11.** *Let E be a BCK/BCI-algebra. Let $\mathcal{F}_A \in S(U)$ for a given subalgebra A of E. Then, $\mathcal{F}_A$ is called an $(\alpha, \beta)$-US algebra of A over U if, for all $x, y \in A$, it satisfies the condition:*

$$\mathcal{F}_A(x * y) \cap \beta \subseteq \mathcal{F}_A(x) \cup \mathcal{F}_A(y) \cup \alpha.$$

We consider the pre-order relation "$\subseteq''_{(\alpha,\beta)}$" on $S(U)$ as: for any $\mathcal{F}_E, \mathcal{G}_E \in S(U)$ and $\varnothing \subseteq \alpha \subset \beta \subseteq U$, we define $\mathcal{F}_E \cap \beta \subseteq \mathcal{G}_E \cup \alpha \Leftrightarrow \mathcal{F}_E(x) \cap \beta \subseteq \mathcal{G}_E(x) \cup \alpha$ for any $x \in E$. We define a relation " $=''_{(\alpha,\beta)}$" such as $\Leftrightarrow \mathcal{F}_E \cap \beta \subseteq \mathcal{G}_E \cup \alpha$ and $\mathcal{G}_E \cap \beta \subseteq \mathcal{F}_E \cup \alpha$. Using the above notion, the $(\alpha, \beta)$-US *BCK/BCI*-algebra is defined as follows:

**Definition 12.** *Let E be a BCK/BCI-algebra. Let $\mathcal{F}_A \in S(U)$ for a given subalgebra A of E. Then, $\mathcal{F}_A$ is called an $(\alpha, \beta)$-US algebra of A over U if, for all $x, y \in A$, it satisfies the condition:*

$$\mathcal{F}_A(x * y) \cap \beta \subseteq \mathcal{F}_A(x) \cup \mathcal{F}_A(y) \cup \alpha.$$

**Example 4.** *Let $X = \{0, a, b, c, d\}$ be a BCK-algebra with the following Cayley table:*

| $*$ | 0 | $a$ | $b$ | $c$ | $d$ |
|---|---|---|---|---|---|
| 0 | 0 | 0 | 0 | 0 | 0 |
| $a$ | $a$ | 0 | 0 | 0 | 0 |
| $b$ | $b$ | $b$ | 0 | 0 | 0 |
| $c$ | $c$ | $c$ | $c$ | 0 | 0 |
| $d$ | $d$ | $c$ | $c$ | $a$ | 0 |

*Let $(\mathcal{F}_A, A)$ be a soft set over $U = X$, where $E = A = X$ and $\mathcal{F}_A : A \to \mathcal{P}(U)$ is a set-valued function defined by $\mathcal{F}_A(x) = \{y \in X | y * x = 0\}$ for all $x \in A$. Then, $\mathcal{F}_A(0) = \{0\}$, $\mathcal{F}_A(a) = \{0, a\}$, $\mathcal{F}_A(b) = \{0, a, b\}$, $\mathcal{F}_A(c) = \{0, a, b, c\}$, and $\mathcal{F}_A(d) = \{0, a, b, c, d\}$. It can be easily verified that $\mathcal{F}_A$ is an $(\alpha, \beta)$-US algebra of A over U, where $\beta = \{0, a, c, d\}$ and $\alpha = \{0, a, d\}$.*

**Theorem 2.** *Let E be a BCK/BCI-algebra, $\mathcal{F}_A \in S(U)$ be a given subalgebra A of E, and $\beta \subseteq U$. For $\delta \in U$, $\mathcal{F}_A$ is an $(\alpha, \beta)$-US subalgebra of A over U if and only if each non-empty subset $B(\mathcal{F}_A : \delta)$, which is defined by:*

$$B(\mathcal{F}_A : \delta) = \{x \in A | \mathcal{F}_A(x) \subseteq \delta \cup \alpha\}$$

*where $\delta \subseteq \beta$, is a subalgebra of A.*

**Proof.** Let $\mathcal{F}_A$ be an $(\alpha, \beta)$-US algebra of $A$ over $U$ such that $\mathcal{F}_A(x) \subseteq \beta$ for every $x \in A$, and let $x, y \in B(\mathcal{F}_A : \delta)$. Then, $\mathcal{F}_A(x * y) \cap \beta \subseteq \mathcal{F}_A(x) \cup \mathcal{F}_A(y) \cup \alpha \subseteq \delta \cup \alpha$, which implies that $x * y \in B(\mathcal{F}_A : \delta)$. Hence, $B(\mathcal{F}_A : \delta)$ is a subalgebra of $A$.

Conversely, let each non-empty subset $B(\mathcal{F}_A : \delta)$ be a subalgebra of $A$. Then, according to our assumption on $\mathcal{F}_A$, for $x, y \in A$, there are $\delta_1, \delta_2 \subseteq \beta$ such that $\mathcal{F}_A(x) = \delta_1$ and $\mathcal{F}_A(y) = \delta_2$. Thus, $\mathcal{F}_A(x) \subseteq \delta$ and $\mathcal{F}_A(y) \subseteq \delta$ for $\delta = \delta_1 \cup \delta_2 \subseteq \beta$. Hence, $x, y \in B(\mathcal{F}_A : \delta)$. Since $B(\mathcal{F}_A : \delta)$ is a subalgebra of $A$, so $x * y \in B(\mathcal{F}_A : \delta)$. Thus, $\mathcal{F}_A(x * y) \cap \beta \subseteq \delta$ and $\mathcal{F}_A(x) \cup \mathcal{F}_A(y) \cup \alpha = \delta_1 \cup \delta_2 \cup \alpha = \delta \cup \alpha$, which implies $\mathcal{F}_A(x * y) \cap \beta \subseteq \mathcal{F}_A(x) \cup \mathcal{F}_A(y) \cup \alpha$. Hence, the proof of the theorem is completed. $\square$

**Theorem 3.** *Let E be a BCK/BCI-algebra and $\mathcal{F}_A \in S(U)$ be such that $A \subseteq E$. Then, $\mathcal{F}_A$ is an $(\alpha, \beta)$-US algebra of A over U if, for all $x \in A$, it satisfies the condition:*

$$\mathcal{F}_A(0) \cap \beta \subseteq \mathcal{F}_A(x) \cup \alpha.$$

**Proof.** If $0 \notin A$, then $\mathcal{F}_A(0) \cap \beta = \emptyset \cap \beta \subseteq \mathcal{F}_A(x) \cup \alpha$ for all $x \in A$. If $0 \in A$, then $\mathcal{F}_A(0) \cap \beta = \mathcal{F}_A(x * x) \cap \beta \subseteq \mathcal{F}_A(x) \cup \mathcal{F}_A(x) \cup \alpha = \mathcal{F}_A(x) \cup \alpha$ for all $x \in A$. Therefore, $\mathcal{F}_A(0) \cap \beta \subseteq \mathcal{F}_A(x) \cup \alpha$ holds. □

**Theorem 4.** *If a soft $\mathcal{F}_A$ over U is an $(\alpha, \beta)$-US algebra of A, then:*

$$(\mathcal{F}_A(0) \cap \beta) \cup \alpha \subseteq (\mathcal{F}_A(x) \cap \beta) \cup \alpha, \text{ for all } x \in A.$$

**Proof.** Let $\mathcal{F}_A \in S(U)$, and by using Theorem 3, we get:

$$
\begin{aligned}
(\mathcal{F}_A(0) \cap \beta) \cup \alpha &= (\mathcal{F}_A(x * x) \cap \beta) \cup \alpha \\
&\subseteq ((\mathcal{F}_A(x) \cup \mathcal{F}_A(x) \cup \alpha) \cap \beta) \cup \alpha \\
&= ((\mathcal{F}_A(x) \cap \beta) \cup \alpha) \cup ((\mathcal{F}_A(x) \cap \beta) \cup \alpha) \\
&\subseteq (\mathcal{F}_A(x) \cap \beta) \cup \alpha.
\end{aligned}
$$

The proof of the theorem is complete. □

**Theorem 5.** *Let E be a BCI-algebra and $\mathcal{F}_A \in S(U)$ for a given subalgebra A of E. Then, $\mathcal{F}_A$ is an $(\alpha, \beta)$-US algebra of A over U if, for all $x \in A$, it satisfies the condition:*

$$\mathcal{F}_A(x * (0 * y)) \cap \beta \subseteq \mathcal{F}_A(x) \cup \mathcal{F}_A(y) \cup \alpha.$$

**Proof.** By using Theorem 3, we have:

$$
\mathcal{F}_A(x * (0 * y)) \cap \beta \subseteq \mathcal{F}_A(x) \cup \mathcal{F}_A(0 * y) \cup \alpha
$$
$$
\subseteq \mathcal{F}_A(x) \cup \mathcal{F}_A(0) \cup \mathcal{F}_A(y) \cup \alpha = \mathcal{F}_A(x) \cup \mathcal{F}_A(y) \cup \alpha.
$$

Therefore, $\mathcal{F}_A(x * (0 * y)) \cap \beta \subseteq \mathcal{F}_A(x) \cup \mathcal{F}_A(y) \cup \alpha$ holds for all $x, y \in A$. □

**Proposition 2.** *Let E be a BCK/BCI-algebra and $\mathcal{F}_A \in S(U)$ for a given subalgebra A of E. Then, $\mathcal{F}_A$ is a $(\alpha, \beta)$-US algebra of A over U if for all $x \in A$, it satisfies the condition:*

$$\mathcal{F}_A(x * y) \cap \beta \subseteq \mathcal{F}_A(y) \cup \alpha \Leftrightarrow \mathcal{F}_A(x) \cap \beta = \mathcal{F}_A(y) \cup \alpha.$$

**Proof.** We assume that $\mathcal{F}_A(x * y) \cap \beta \subseteq \mathcal{F}_A(y) \cup \alpha$ for all $x, y \in A$. Take $y = 0$, and use $(C_6)$, which induces $\mathcal{F}_A(x) \cap \beta = \mathcal{F}_A(x * 0) \cap \beta \subseteq \mathcal{F}_A(0) \cup \alpha$. It follows from Theorem 3 that $\mathcal{F}_A(x) \cap \beta = \mathcal{F}_A(0) \cup \alpha$ for all $x \in A$.

Conversely, suppose that $\mathcal{F}_A(x) \cap \beta = \mathcal{F}_A(0) \cup \alpha$ for all $x \in A$. Then,

$$\mathcal{F}_A(x * y) \cap \beta \subseteq \mathcal{F}_A(x) \cup \mathcal{F}_A(y) \cup \alpha = \mathcal{F}_A(0) \cup \mathcal{F}_A(y) \cup \alpha = \mathcal{F}_A(y) \cup \alpha$$

for all $x, y \in A$. □

For a soft set $(\mathcal{F}_A, A)$ over $E$, we consider the set:

$$X_0 = \{x \in A | \mathcal{F}_A(x) = \mathcal{F}_A(0)\}.$$

**Theorem 6.** *Let E be a BCK/BCI-algebra and A a subalgebra of E. Let $(\mathcal{F}_A, A)$ be an $(\alpha, \beta)$-US algebra over E. Then, the set $X_0^* = \{x \in A | (\mathcal{F}_A(x) \cap \beta) \cup \alpha = (\mathcal{F}_A(0) \cap \beta) \cup \alpha\}$ is a subalgebra of E.*

**Proof.** If $\mathcal{F}_A$ is an $(\alpha, \beta)$-US algebra of $A$ over $U$, then $x, y \in X_0^*$; we have $(\mathcal{F}_A(x) \cap \beta) \cup \alpha = (\mathcal{F}_A(0) \cap \beta) \cup \alpha = (\mathcal{F}_A(y) \cap \beta) \cup \alpha$. Then, from Theorem 3, we have $(\mathcal{F}_A(0) \cap \beta) \cup \alpha \subseteq (\mathcal{F}_A(x * y) \cap \beta) \cup \alpha$ for all $x, y \in A$. This also takes the following form, $(\mathcal{F}_A(x * y) \cap \beta) \cup \alpha \subseteq ((\mathcal{F}_A(x) \cup \mathcal{F}_A(y) \cup \alpha) \cap \beta) \cup \alpha = ((\mathcal{F}_A(x) \cap \beta) \cup \alpha) \cup (\mathcal{F}_A(y) \cap \beta) \cup \alpha) \subseteq (\mathcal{F}_A(0) \cap \beta) \cup \alpha$. Hence, $(\mathcal{F}_A(x * y) \cap \beta) \cup \alpha = (\mathcal{F}_A(0) \cap \beta) \cup \alpha$, and so, $x * y \in X_0^*$. Thus, $X_0^*$ is a subalgebra of $A$. $\square$

**Theorem 7.** *Let $E$ be a BCK-algebra and $\mathcal{F}_A \in S(U)$. Define a soft set $\mathcal{F}_A^*$ over $U$ by $\mathcal{F}_A^* : E \to \mathcal{P}(U)$,*

$$x \longmapsto \begin{cases} \mathcal{F}_A(x) & \text{if } x \in B(\mathcal{F}_A : \delta) \\ U & \text{otherwise.} \end{cases}$$

*If $\mathcal{F}_A$ is an $(\alpha, \beta)$-US algebra over $U$, then so is $\mathcal{F}_A^*$.*

**Proof.** If $\mathcal{F}_A$ is an $(\alpha, \beta)$-US algebra over $U$, then $B(\mathcal{F}_A : \delta)$ is a subalgebra of $A$ by Theorem 2. Let $x, y \in A$. If $x, y \in B(\mathcal{F}_A : \delta)$, then $x * y \in B(\mathcal{F}_A : \delta)$, and so,

$$\mathcal{F}_A^*(x * y) \cap \beta = \mathcal{F}_A(x * y) \cap \beta \subseteq \mathcal{F}_A(x) \cup \mathcal{F}_A(y) \cup \alpha = \mathcal{F}_A^*(x) \cup \mathcal{F}_A^*(y) \cup \alpha.$$

If $x \notin B(\mathcal{F}_A : \delta)$ or $y \notin B(\mathcal{F}_A : \delta)$, then $\mathcal{F}_A^*(x) = U$ or $\mathcal{F}_A^*(y) = U$. Thus, we have:

$$\mathcal{F}_A^*(x * y) \cap \beta \subseteq U = \mathcal{F}_A^*(x) \cup \mathcal{F}_A^*(y) \cup \alpha.$$

Therefore, $\mathcal{F}_A^*$ is an $(\alpha, \beta)$-US algebra of $A$ over $U$. $\square$

## 5. $(\alpha, \beta)$-US Ideals in *BCK/BCI*-Algebras

In this section, we define the $(\alpha, \beta)$-US ideal and $(\alpha, \beta)$-US closed ideal and characterize their properties in detail.

**Definition 13.** *Let $E$ be a BCK/BCI-algebra and $A$ be a subalgebra of $E$. Let $\mathcal{F}_A \in S(U)$, then $\mathcal{F}_A$ is called an $(\alpha, \beta)$-US ideal over $U$ if, for all $x, y \in A$, it satisfies Theorem 3 and the following condition:*
*$\mathcal{F}_A(x) \cap \beta \subseteq \mathcal{F}_A(x * y) \cup \mathcal{F}_A(y) \cup \alpha.$*

**Example 5.** *Let $U = Z$ (set of positive integers) be the universal set and $E = \{0, a, b, c, d\}$ be a BCK-algebra with the following Cayley table:*

| $*$ | 0 | $a$ | $b$ | $c$ | $d$ |
|---|---|---|---|---|---|
| 0 | 0 | 0 | 0 | 0 | 0 |
| $a$ | $a$ | 0 | $a$ | 0 | $a$ |
| $b$ | $b$ | $b$ | 0 | 0 | 0 |
| $c$ | $c$ | $c$ | $c$ | 0 | $c$ |
| $d$ | $d$ | $d$ | $b$ | $b$ | 0 |

*For a subalgebra $A = \{0, b, c, d\}$ of $E$, define the soft set $(\mathcal{F}_A, A)$ over $U$ as $\mathcal{F}_A(0) = \{1, 3, 4, 5, 7, 9, 11, 12\}$, $\mathcal{F}_A(b) = \{1, 2, 4, 5, 6, 7, 8, 10, 13\}$, $\mathcal{F}_A(c) = \{2, 3, 5, 6, 8, 9, 13\}$, and $\mathcal{F}_A(d) = \{1, 2, 3, 5, 8, 10, 13\}$. Then, $\mathcal{F}_A$ is an $(\alpha, \beta)$-US ideal of $A$ over $U$ where $\beta = \{1, 2, 3, 6, 7, 8, 9, 10, 11, 12, 13\}$ and $\alpha = \{1, 3, 6, 7, 9, 10, 11, 12\}$.*

**Example 6.** *Let $U = N$ (set of natural numbers) be the universal set and $E = \{0, a, b, c, d\}$ be a BCK-algebra with the following Cayley table:*

| * | 0 | a | b | c | d |
|---|---|---|---|---|---|
| 0 | 0 | 0 | 0 | 0 | 0 |
| a | a | 0 | 0 | 0 | 0 |
| b | b | b | 0 | 0 | 0 |
| c | c | c | c | 0 | 0 |
| d | d | c | c | a | 0 |

*Define the soft set $\mathcal{F}_A$ over U as follows $\mathcal{F}_A(0) = N$, $\mathcal{F}_A(a) = 2N$, $\mathcal{F}_A(b) = 4N$, $\mathcal{F}_A(c) = 6N$, and $\mathcal{F}_A(d) = 8N$. Then, $\mathcal{F}_A$ is not an $(\alpha, \beta)$-US ideal over U, where $\beta = 12N$ and $\alpha = 24N$, because:*

$$\mathcal{F}_A(0) \cap \beta = 12N \nsubseteq 8N = \mathcal{F}_A(d) \cup \alpha.$$

**Lemma 1** ([42])**.** *Let E be a BCK/BCI-algebra and A be a subalgebra of E. Let $\mathcal{F}_A \in S(U)$, if $\mathcal{F}_A$ is a US ideal over U, then for all $x, y \in A$:*

$$x \leq y \Rightarrow \mathcal{F}_A(x) \subseteq \mathcal{F}_A(y).$$

**Proof.** Let $x, y \in A$ be such that $x \leq y$. Then, $x * y = 0$, from which, by Definition 13 and Theorem 3, we get $\mathcal{F}_A(x) \subseteq \mathcal{F}_A(x * y) \cup \mathcal{F}_A(y) = \mathcal{F}_A(0) \cup \mathcal{F}_A(y) = \mathcal{F}_A(y)$. Hence, $\mathcal{F}_A(x) \subseteq \mathcal{F}_A(y)$.  □

**Lemma 2.** *Let E be a BCK/BCI-algebra and A be a subalgebra of E. Let $\mathcal{F}_A \in S(U)$. If $\mathcal{F}_A$ is an $(\alpha, \beta)$-US ideal over U, then for all $x, y \in A$:*

$$x \leq y \Rightarrow \mathcal{F}_A(x) \cap \beta \subseteq \mathcal{F}_A(y) \cup \alpha.$$

**Proof.** Let $x, y \in A$ be such that $x \leq y$. Then, $x * y = 0$, from which, by Definition 13 and Theorem 3, we get $\mathcal{F}_A(x) \cap \beta \subseteq \mathcal{F}_A(x * y) \cup \mathcal{F}_A(y) \cup \alpha = \mathcal{F}_A(0) \cup \mathcal{F}_A(y) \cup \alpha = \mathcal{F}_A(y) \cup \alpha$. Hence, $\mathcal{F}_A(x) \cap \beta \subseteq \mathcal{F}_A(y) \cup \alpha$.  □

**Proposition 3.** *Let E be a BCK/BCI-algebra. For a given subalgebra A of E, let $\mathcal{F}_A \in S(U)$. If $\mathcal{F}_A$ is an $(\alpha, \beta)$-US ideal over U, then for all $x, y, z \in A$, $\mathcal{F}_A$ satisfies the following conditions:*
*(1) $\mathcal{F}_A(x * y) \cap \beta \subseteq \mathcal{F}_A(x * z) \cup \mathcal{F}_A(z * y) \cup \alpha$*
*(2) $\mathcal{F}_A(x * y) = \mathcal{F}_A(0) \Rightarrow \mathcal{F}_A(x) \cap \beta \subseteq \mathcal{F}_A(y) \cup \alpha$.*

**Proof.** (1) Since $(x * y) * (x * z) \leq z * y$, then from Lemma 2, $\mathcal{F}_A((x * y) * (x * z)) \subseteq \mathcal{F}_A(z * y)$. Hence,

$$\mathcal{F}_A(x * y) \cap \beta \subseteq \mathcal{F}_A((x * y) * (x * z)) \cup \mathcal{F}_A(x * z) \cup \alpha \subseteq \mathcal{F}_A(x * z) \cup \mathcal{F}_A(z * y) \cup \alpha.$$

(2) If $\mathcal{F}_A(x * y) = \mathcal{F}_A(0)$, then for all $x, y \in A$,

$$\mathcal{F}_A(x) \cap \beta \subseteq \mathcal{F}_A(x * y) \cup \mathcal{F}_A(y) \cup \alpha = \mathcal{F}_A(0) \cup \mathcal{F}_A(y) \cup \alpha = \mathcal{F}_A(y) \cup \alpha.$$

□

**Proposition 4.** *Let E be a BCK/BCI-algebra and A be a subalgebra of E. If $\mathcal{F}_A$ is an $(\alpha, \beta)$-US ideal over U, then for all $x, y, z \in A$, the following conditions are equivalent:*
*(1) $\mathcal{F}_A(x * y) \cap \beta \subseteq \mathcal{F}_A((x * y) * y) \cup \alpha$.*
*(2) $\mathcal{F}_A((x * z) * (y * z)) \cap \beta \subseteq \mathcal{F}_A((x * y) * z) \cup \alpha$.*

**Proof.** Assume that (1) holds and $x, y, z \in A$. Since $((x * (y * z)) * z) * z = ((x * z) * (y * z)) * z \leq (x * y) * z$ by (1), $(C_8)$, and Lemma 2, we obtain the following equality: $\mathcal{F}_A((x * z) * (y * z)) \cap \beta = \mathcal{F}_A((x * (y * z)) * z) \cap \beta \subseteq \mathcal{F}_A(((x * (y * z)) * z) * z) \cup \alpha \subseteq \mathcal{F}_A((x * y) * z) \cup \alpha$.

Again, assume that (2) holds. If we put $y = z$ in (2), then by $(C_3)$ and $(C_6)$, we get $\mathcal{F}_A((x * z) * z) \cup \alpha \supseteq \mathcal{F}_A((x * z) * (z * z)) \cap \beta = \mathcal{F}_A((x * z) * 0) \cap \beta = \mathcal{F}_A(x * z) \cap \beta$, which implies that (1) holds. □

**Theorem 8.** *Let E be a BCK/BCI-algebra and A be a given subalgebra of E. Then, every A-soft set is an $(\alpha, \beta)$-US ideal over U, and an A-soft set is an $(\alpha, \beta)$-US BCK/BCI-algebra over U.*

**Proof.** Let $\mathcal{F}_A$ be an $(\alpha, \beta)$-US ideal over $U$ and $A$ a subalgebra of $E$. From [42], we get $x * y \leq x$ for all $x, y \in A$. Then, it follows from Lemma 2 that $\mathcal{F}_A(x * y) \cap \beta \subseteq \mathcal{F}_A(x) \cup \alpha \subseteq \mathcal{F}_A(x * y) \cup \mathcal{F}_A(y) \cup \alpha \subseteq \mathcal{F}_A(x) \cup \mathcal{F}_A(y) \cup \alpha$. Hence, $\mathcal{F}_A$ is an $(\alpha, \beta)$-US $BCK/BCI$-algebra over $U$. □

The converse of Theorem 8 is not true. This is justified by the following example.

**Example 7.** *Let $U = N$ be the initial universal set. Let $E = N$ be the set of natural numbers, and define a binary operation $*$ on E such that:*

$$x * y = \frac{x}{(x, y)}$$

*for all $x, y \in X$, where $(x, y)$ is the greatest common divisor of x and y. Then, $(X; *, 1)$ is a BCK-algebra. For a subalgebra $A = \{1, 2, 3, 4, 5\}$ of E, define the soft set $(\mathcal{F}_A, A)$ over U as follows: $\mathcal{F}_A(1) = N$, $\mathcal{F}_A(2) = 4N$, $\mathcal{F}_A(3) = 2N$, $\mathcal{F}_A(4) = 3N$, and $\mathcal{F}_A(5) = 8N$. Then, $\mathcal{F}_A$ is an $(\alpha, \beta)$-US algebra of A over U, but it is not an $(\alpha, \beta)$-US ideal of A over U, where $\beta = 6N$ and $\alpha = 12N$, because:*

$$\mathcal{F}_A(4) \cap \beta = 18N \nsubseteq 4N = \mathcal{F}_A(4 * 2) \cup \mathcal{F}_A(2) \cup \alpha$$

**Theorem 9.** *Let E be a BCK/BCI-algebra. Let $\mathcal{F}_A \in S(U)$ and A be a subalgebra of E. If $\mathcal{F}_A$ is an $(\alpha, \beta)$-US ideal over U, then for all $x, y, z \in A$, $\mathcal{F}_A$ satisfies the following condition:*

$$x * y \leq z \Rightarrow \mathcal{F}_A(x) \cap \beta \subseteq \mathcal{F}_A(y) \cup \mathcal{F}_A(z) \cup \alpha.$$

**Proof.** Let $x, y \in A$ be such that $x * y \leq z$, then $(x * y) * z = 0 \Rightarrow$

$$\mathcal{F}_A(x * y) \cap \beta \subseteq \mathcal{F}_A((x * y) * z) \cup \mathcal{F}_A(z) \cup \alpha = \mathcal{F}_A(0) \cup \mathcal{F}_A(z) \cup \alpha = \mathcal{F}(z) \cup \alpha.$$

Also, from which by using Definition 13 and Theorem 3, follows as $\mathcal{F}_A(x) \cap \beta \subseteq \mathcal{F}_A(x * y) \cup \mathcal{F}_A(y) \subseteq \mathcal{F}_A(y) \cup \mathcal{F}_A(z) \cup \alpha$. □

The following results can be proven by induction.

**Corollary 1.** *Let E be a BCK/BCI-algebra and A be a subalgebra of E. Let $\mathcal{F}_A \in S(U)$, which satisfies the hypothesis of Theorem 3. Then, $\mathcal{F}_A$ is an $(\alpha, \beta)$-US ideal over U if and only if for all $x, a_1, a_2, \ldots, a_n \in A$, it satisfies the following condition:*

$$x * \prod_{i=1}^{n} a_i = 0 \Rightarrow \mathcal{F}_A(x) \cap \beta \subseteq \bigcup_{i=1,2,\ldots,n} \mathcal{F}_A(a_i) \cup \alpha.$$

We establish the following lemmas.

**Lemma 3.** *Let E be a BCK-algebra such that for all $x, a, b, a_1, a_2, \ldots, a_n, b_1, b_2, \ldots, b_m \in E$, the three conditions $(x * a) * b = 0$, $a * \prod_{i=1}^{n} a_i = 0$ and $b * \prod_{j=1}^{m} b_j = 0$ are satisfied. Then, $(x * \prod_{i=1}^{n} a_i) * \prod_{j=1}^{m} b_j = 0$.*

**Proof.** From $(x * a) * b = 0$, it follows that $x * a \leq b$.

Successively $*$-multiplying the above inequality on the right-hand side by $a_1, a_2, \ldots, a_n$ gives $(x * \prod_{i=1}^{n} a_i) * b = 0$; thus, $x * \prod_{i=1}^{n} a_i \leq b$.

Then, successively multiplying the right-hand side of the above inequality by $b_1, b_2, \ldots, b_m$ gives $(x * \prod_{i=1}^{n} a_i) * \prod_{j=1}^{m} b_j \leq b * \prod_{j=1}^{m} b_j = 0$.

Thus, $(x * \prod_{i=1}^{n} a_i) * \prod_{j=1}^{m} b_j = 0$. This completes the proof.　□

**Lemma 4.** *Let E be a BCK-algebra satisfying the three conditions of Lemma 3. If $\mathcal{F}_A$ is an $(\alpha, \beta)$-US ideal over U, then:*

$$\mathcal{F}_A(x) \cap \beta \subseteq \bigcup_{i=1,2,\ldots,n, j=1,2,\ldots,m} (\mathcal{F}_A(a_i) \cup \mathcal{F}_A(b_j)) \cup \alpha.$$

**Proof.** This follows from Lemma 3 and Corollary 1.　□

**Theorem 10.** *Let E be a BCK/BCI-algebra. Given a subalgebra A of E, let $\mathcal{F}_A \in S(U)$ and $\beta \subseteq U$. Then, $\mathcal{F}_A$ is an $(\alpha, \beta)$-US ideal over U if and only if the non-empty set $B(\mathcal{F}_A : \delta)$ is an ideal of A.*

**Proof.** The proof of the theorem is the same as Theorem 2.　□

**Definition 14.** *Let E be a BCK/BCI-algebra. For a given subalgebra A of E, let $\mathcal{F}_A \in S(U)$. An $(\alpha, \beta)$-US ideal $\mathcal{F}_A$ is said to be closed if for all $x \in A$, it satisfies the condition:*

$$\mathcal{F}_A(0 * x) \cap \beta \subseteq \mathcal{F}_A(x) \cup \alpha.$$

**Example 8.** *Let $U = Z$ (set of positive integers) be the universal set and $E = \{0, 1, 2, a, b\}$ be a BCI-algebra with the following Cayley table:*

| $*$ | 0 | 1 | 2 | $a$ | $b$ |
|---|---|---|---|---|---|
| 0 | 0 | 0 | 0 | $a$ | $a$ |
| 1 | 1 | 0 | 1 | $b$ | $a$ |
| 2 | 2 | 2 | 0 | $a$ | $a$ |
| $a$ | $a$ | $a$ | $a$ | 0 | 0 |
| $b$ | $b$ | $a$ | $b$ | 1 | 0 |

*For a subalgebra $A = \{0, 1, 2, a, b\}$ of E, define the soft set $(\mathcal{F}_A, A)$ over U as $\mathcal{F}_A(0) = \{1, 3, 4, 5, 7, 9\}$, $\mathcal{F}_A(1) = \{1, 2, 4, 5, 6, 7, 8, 10\}$, $\mathcal{F}_A(2) = \{1, 2, 4, 5, 6, 8, 12, 13\}$, $\mathcal{F}_A(a) = \{2, 3, 5, 6, 8, 9, 12, 13\}$, $\mathcal{F}_A(b) = \{1, 2, 3, 5, 7, 10, 12, 13\}$, $\beta = \{1, 2, 3, 5, 7, 9, 10, 11, 12, 13\}$, and $\alpha = \{1, 3, 5, 6, 7, 8, 9, 11\}$. Then, $\mathcal{F}_A$ is an $(\alpha, \beta)$-US closed ideal of A over U.*

**Theorem 11.** *Let E be a BCI-algebra. Then, an $(\alpha, \beta)$-US ideal over U is closed if and only if it is an $(\alpha, \beta)$-US algebra over U.*

**Proof.** Let $\mathcal{F}_A$ be an $(\alpha, \beta)$-US ideal over U. If $\mathcal{F}_A$ is closed, then $\mathcal{F}_A(0 * x) \cap \beta \subseteq \mathcal{F}_A(x) \cup \alpha$, for all $x \in A$. It follows from Definition 13 that:

$$\mathcal{F}_A(x * y) \cap \beta \subseteq \mathcal{F}_A((x * y) * x) \cup \mathcal{F}_A(x) \cup \alpha = \mathcal{F}_A(0 * y) \cup \mathcal{F}_A(x) \cup \alpha \subseteq \mathcal{F}_A(x) \cup \mathcal{F}_A(y) \cup \alpha$$

for all $x, y \in A$. Hence, $\mathcal{F}_A$ is an $(\alpha, \beta)$-US algebra of A over U.

Conversely, if $\mathcal{F}_A$ is an $(\alpha, \beta)$-US algebra over U, then,

$$\mathcal{F}_A(0 * x) \cap \beta \subseteq \mathcal{F}_A(0) \cup \mathcal{F}_A(x) \cup \alpha = \mathcal{F}_A(x) \cup \alpha.$$

for all $x \in A$. Thus, $\mathcal{F}_A$ is an $(\alpha, \beta)$-US closed ideal over $U$. $\quad\square$

Let $E$ be a $BCI$-algebra and $B(E) = \{x \in E | 0 \leq x\}$. For any $x \in E$ and $n \in N$ ($N$ is the set of natural numbers), define $x^n$ by:

$$x^1 = x, x^{n+1} = x * (0 * x).$$

If there exists an $n \in N$ such that $x^n \in B(E)$, then we say that $x$ is finite periodic (see [50]), and its period is denoted by $|x|$ and defined by:

$$|x| = \min\{n \in N | x^n \in B(E)\}.$$

Otherwise, $x$ is infinite ordered and denoted by $|x| = infinite$.

**Theorem 12.** *Let $E$ be a BCI-algebra in which every element is of finite period. Then, every $(\alpha, \beta)$-US ideal over $U$ is closed.*

**Proof.** Let $\mathcal{F}_E$ be an $(\alpha, \beta)$-US ideal over $U$. Then, for any $x \in E$, suppose that $|x| = n$. Then, $x^n \in B(X)$. We get $(0 * x^{n-1}) * x = (0 * (0 * (0 * x^{n-1}))) * x = (0 * x) * (0 * (0 * x^{n-1})) = 0 * (x * (0 * x^{n-1})) = 0 * x^n = 0$, and so, $\mathcal{F}_E((0 * x^{n-1}) * x) = \mathcal{F}_E(0) \subseteq \mathcal{F}_E(x)$ by using Theorem 3. Then, from Definition 13, it follows that:

$$\mathcal{F}_E((0 * x^{n-1}) * x) \cap \beta \subseteq \mathcal{F}_E(0 * x^{n-1}) \cup \mathcal{F}_E(x) \cup \alpha = \mathcal{F}_E(0) \cup \mathcal{F}_E(x) \cup \alpha = \mathcal{F}_E(x) \cup \alpha.$$

Furthermore, it is noted that $(0 * x^{n-2}) * x = (0 * (0 * (0 * x^{n-2}))) * x = (0 * x) * (0 * (0 * x^{n-2})) = 0 * (x * (0 * x^{n-2})) = 0 * x^{n-1}$, which implies from above that:

$$\mathcal{F}_E((0 * x^{n-2}) * x) = \mathcal{F}_E(0 * x^{n-1}) \subseteq \mathcal{F}_E(x).$$

Again, by Definition 13, we have:

$$\mathcal{F}_E((0 * x^{n-2}) * x) \cap \beta \subseteq \mathcal{F}_E(0 * x^{n-2}) \cup \mathcal{F}_E(x) \cup \alpha = \mathcal{F}_E(0) \cup \mathcal{F}_E(x) \subseteq \mathcal{F}_E(x) \cup \alpha.$$

By continuation of the above process, we get $\mathcal{F}_E(0 * x) \cap \beta \subseteq \mathcal{F}_E(x) \cup \alpha$ for all $x \in E$. Hence, $\mathcal{F}_E$ is an $(\alpha, \beta)$-US closed ideal over $U$. $\quad\square$

## 6. $(\alpha, \beta)$-US Commutative Ideals in *BCK/BCI*-Algebras

**Definition 15.** *Let $E$ be a BCK-algebra. For a given subalgebras $A$ of $E$, let $\mathcal{F}_A \in S(U)$. Then, $\mathcal{F}_A$ is called an $(\alpha, \beta)$-US commutative ideal over $U$ if for all $x, y, z \in A$, it satisfies Theorem 3 and the following condition:*

$$\mathcal{F}_A(x * (y * (y * x))) \cap \beta \subseteq \mathcal{F}_A((x * y) * z) \cup \mathcal{F}_A(z) \cup \alpha.$$

**Example 9.** *Let $E = \{0, 1, 2, 3, 4\}$ be a BCK-algebra with the following Cayley table:*

| * | 0 | 1 | 2 | 3 | 4 |
|---|---|---|---|---|---|
| 0 | 0 | 0 | 0 | 0 | 0 |
| 1 | 1 | 0 | 1 | 1 | 0 |
| 2 | 2 | 2 | 0 | 2 | 0 |
| 3 | 3 | 3 | 3 | 0 | 0 |
| 4 | 4 | 4 | 4 | 4 | 0 |

*Let $(\mathcal{F}_A, A)$ be a soft set over $U = X$, where $A = \{1, 2, 3, 4\}$ and $\mathcal{F}_A : A \to P(X)$ is a set valued function defined by $\mathcal{F}_A(x) = \{y \in X | y * x \in \{0, 2, 3\}\}$. Then,*

$$\mathcal{F}_A(0) = \varnothing,$$

$$\mathcal{F}_A(1) = \{y \in X \mid y * 1 \in \{0, 2, 3\}\} = \{0, 1, 2, 3\},$$

$$\mathcal{F}_A(2) = \{y \in X \mid y * 2 \in \{0, 2, 3\}\} = \{0, 2, 3\},$$

$$\mathcal{F}_A(3) = \{y \in X \mid y * 3 \in \{0, 2, 3\}\} = \{0, 2, 3\},$$

$$\mathcal{F}_A(4) = \{y \in X \mid y * 4 \in \{0, 2, 3\}\} = \{0, 1, 2, 3, 4\}.$$

Then, $\mathcal{F}_A$ is an $(\alpha, \beta)$-US commutative ideal of $A$ over $U$, where $\beta = \{0, 1, 3, 4\}$ and $\alpha = \{0, 1, 3\}$.

**Theorem 13.** *Let $E$ be a BCK-algebra. Then, any $(\alpha, \beta)$-US commutative ideal over $U$ is an $(\alpha, \beta)$-US ideal over $U$.*

**Proof.** Let $A$ be a subalgebra of $E$ and $\mathcal{F}_A$ be an $(\alpha, \beta)$-US commutative ideal over $U$. Now, we put $y = 0$ in Definition 15 and use $(C_5)$ and $(C_6)$, then we have $\mathcal{F}_A(x) \cap \beta = \mathcal{F}_A(x * (0 * (0 * x))) \cap \beta \subseteq \mathcal{F}_A((x * 0) * z) \cup \mathcal{F}_A(z) \cup \alpha = \mathcal{F}_A(x * z) \cup \mathcal{F}_A(z) \cup \alpha$ for all $x, z \in A$. Thus, $\mathcal{F}_A$ is an $(\alpha, \beta)$-US ideal over $U$. $\square$

In view of the following example, we can also establish Theorem 13.

**Example 10.** *Let $U = N$ (set of natural numbers) be the universal set and $E = \{0, a, b, c, d\}$ be a BCK-algebra with the following Cayley table:*

| * | 0 | a | b | c | d |
|---|---|---|---|---|---|
| 0 | 0 | 0 | 0 | 0 | 0 |
| a | a | 0 | a | 0 | 0 |
| b | b | b | 0 | 0 | 0 |
| c | c | c | c | 0 | 0 |
| d | d | d | d | c | 0 |

*The soft set $(\mathcal{F}_A, A)$ is defined over $U$ as follows $\mathcal{F}_A(0) = N$, $\mathcal{F}_A(a) = 3N$, $\mathcal{F}_A(b) = \mathcal{F}_A(d) = 2N$, and $\mathcal{F}_A(c) = \emptyset$. Then, $\mathcal{F}_A$ is an $(\alpha, \beta)$-US commutative ideal over $U$, as well as an $(\alpha, \beta)$-US ideal over $U$, where $\beta = 6N$ and $\alpha = 12N$.*

The following theorem provides the condition that an $(\alpha, \beta)$-US ideal over $U$ is an $(\alpha, \beta)$-US commutative ideal over $U$.

**Theorem 14.** *Let $E$ be a BCK-algebra and $A$ be a subalgebra of $E$. Let $\mathcal{F}_A \in S(U)$, then $\mathcal{F}_A$ is an $(\alpha, \beta)$-US commutative ideal over $U$ if and only if, for all $x, y, z \in A$, $\mathcal{F}_A$ is an $(\alpha, \beta)$-US ideal over $U$ satisfying the following condition:*

$$\mathcal{F}_A(x * (y * (y * x))) \subseteq \mathcal{F}_A(x * y).$$

**Proof.** Assume that $\mathcal{F}_A$ is an $(\alpha, \beta)$-US ideal commutative ideal over $U$. Then, $\mathcal{F}_A$ is an $(\alpha, \beta)$-US soft ideal over $U$ by Theorem 13. Now, if we take $z = 0$ in Definition 15 and use $(C_5)$, then we deduce the condition given in Theorem 14.

Conversely, if $\mathcal{F}_A$ is an $(\alpha, \beta)$-US ideal over $U$ satisfying the condition of Theorem 14, then for all $x, y, z \in A$, we have $\mathcal{F}_A(x * y) \cap \beta \subseteq \mathcal{F}_A((x * y) * z) \cup \mathcal{F}_A(z) \cup \alpha$ by Definition 13. Hence, from Definition 15, we conclude that $\mathcal{F}_A$ is an $(\alpha, \beta)$-US commutative ideal over $U$. $\square$

**Corollary 2.** *Let $E$ be a BCK-algebra and $\mathcal{F}_E \in S(U)$. Then, $\mathcal{F}_E$ is an $(\alpha, \beta)$-US commutative ideal over $U$ if and only if $\mathcal{F}_E$ is an $(\alpha, \beta)$-US ideal over $U$ satisfying the following condition for all $x, y \in A$:*

$$\mathcal{F}_E(x * (y * (y * x))) \cap \beta \subseteq \mathcal{F}_A(x * y) \cup \alpha.$$

**Theorem 15.** *Let E be a commutative BCK-algebra. Then, every $(\alpha, \beta)$-US ideal over U is an $(\alpha, \beta)$-US commutative ideal over U.*

**Proof.** Let $\mathcal{F}_A$ be an $(\alpha, \beta)$-US ideal over $U$, where $A$ is a subalgebra of $E$. Then, for all $x, y, z \in A$, we notice that:

$$
\begin{aligned}
(((x*(y*(y*x)))*((x*y)*z)))*z &= ((x*(y*(y*x)))*z)*((x*y)*z) \\
&\leq ((x*(y*(y*x)))*(x*y) \\
&= (x*(x*y))*(y*(y*x)) = 0
\end{aligned}
$$

Thus, $(x*(y*(y*x)))*((x*y)*z)) \leq z$. Then, from Theorem 9, we get $\mathcal{F}_A(x*(y*(y*x))) \cap \beta \subseteq \mathcal{F}_A((x*y)*z) \cup \mathcal{F}_A(z) \cup \alpha$. Hence, $\mathcal{F}_A$ is an $(\alpha, \beta)$-US commutative ideal over $U$. $\square$

**Theorem 16.** *Let E be a BCK-algebra and A be a subalgebra of E. Let $\mathcal{F}_A \in S(U)$. If $\mathcal{F}_A$ satisfies the following conditions:*
*(1) $x*(x*y) \leq y*(y*x)$ for all $x, y \in A$;*
*(2) $\mathcal{F}_A$ is an $(\alpha, \beta)$-US ideal over U;*
*then $\mathcal{F}_A$ is an $(\alpha, \beta)$-US commutative ideal over U.*

**Proof.** For any $x, y \in A$, we have:

$$
\mathcal{F}_A(x*(y*(y*x)))*(x*y) = (x*(x*y))*(y*(y*x)) = 0
$$

by $(C_8)$ and (1). Therefore, $x*(y*(y*x)) \leq x*y$ for all $x, y \in A$, which indicates from Lemma 2 that $\mathcal{F}_A(x*(y*(y*x))) \cap \beta \subseteq \mathcal{F}_A(x*y) \cup \alpha$. Now, it follows from Theorem 14 that $\mathcal{F}_A$ is an $(\alpha, \beta)$-US commutative ideal of $A$ over $U$. $\square$

**Theorem 17.** *Let E be a BCK/BCI-algebra and A be a subalgebra of E. Consider $\mathcal{F}_A \in S(U)$ and $\delta \subseteq \beta \subseteq U$. Then, $\mathcal{F}_A$ is an $(\alpha, \beta)$-US commutative ideal over U if and only if the non-empty set $B(\mathcal{F}_A : \delta)$ is a commutative ideal of A.*

**Proof.** The proof of the theorem is the same as Theorem 2. $\square$

**Theorem 18.** *Let E be a BCK-algebra and $\mathcal{F}_A \in S(U)$. Define a soft set $\mathcal{F}_A^*$ over U by $\mathcal{F}_A^* : E \to \mathcal{P}(U)$,*

$$
x \longmapsto \begin{cases} \mathcal{F}_A(x) & \text{if } x \in B(\mathcal{F}_A : \delta) \\ U & \text{otherwise.} \end{cases}
$$

*If $\mathcal{F}_A$ is an $(\alpha, \beta)$-US commutative ideal over U, then so is $\mathcal{F}_A^*$.*

**Proof.** If $\mathcal{F}_A$ is an $(\alpha, \beta)$-US commutative ideal over $U$, then $B(\mathcal{F}_A : \delta)$ is a commutative ideal over $U$ by Theorem 17. Hence, $0 \in B(\mathcal{F}_A : \delta)$, and so, we have $\mathcal{F}_A^*(0) \cap \beta = \mathcal{F}_A(0) \cap \beta \subseteq \mathcal{F}_A(x) \cup \alpha \subseteq \mathcal{F}_A^*(x) \cup \alpha$ for all $x \in A$. Let $x, y, z \in A$. Then, $(x*y)*z \in B(\mathcal{F}_A : \delta)$ and $z \in B(\mathcal{F}_A : \delta)$; hence, $x*(y*(y*x)) \in B(\mathcal{F}_A : \delta)$, and so, we deduce the following equality:

$$
\begin{aligned}
\mathcal{F}_A^*(x*(y*(y*x))) \cap \beta &= \mathcal{F}_A(x*(y*(y*x))) \cap \beta \\
&\subseteq \mathcal{F}_A((x*y)*z) \cup \mathcal{F}_A(z) \cup \alpha \\
&= \mathcal{F}_A^*((x*y)*z) \cup \mathcal{F}_A^*(z) \cup \alpha.
\end{aligned}
$$

If $(x*y)*z \notin B(\mathcal{F}_A : \delta)$ and $z \notin B(\mathcal{F}_A : \delta)$, then $\mathcal{F}_A^*(x*(y*(y*x)))$ or $\mathcal{F}_A^*(z) = U$. Thus, we have $\mathcal{F}_A^*(x*(y*(y*x))) \cap \beta \subseteq U = \mathcal{F}_A^*((x*y)*z) \cup \mathcal{F}_A^*(z) \cup \alpha$. This shows that $\mathcal{F}_A^*$ is an $(\alpha, \beta)$-US commutative ideal of $A$ over $U$. $\square$

**Theorem 19.** *Let E be a BCK-algebra and A be a subset of E, which is a commutative ideal of E if and only if the soft subset $\mathcal{F}_A$ defined by:*

$$\mathcal{F}_A(x) = \begin{cases} \Omega & \text{if } x \in A \\ \Gamma & \text{if } x \notin A, \end{cases}$$

*where $\alpha \subseteq \Omega \subseteq \Gamma \subseteq \beta \subseteq U$, is an $(\alpha, \beta)$-US commutative ideal of A over U.*

**Proof.** Let $A$ be a commutative ideal of $E$ and if $x \in A$, then $0 \in A$. Therefore, $\mathcal{F}_A(0) = \mathcal{F}_A(x) = \Omega$, and so, $\mathcal{F}_A(0) \cap \beta = \Omega \cap \beta = \Omega$ and $\mathcal{F}_A(x) \cup \alpha = \Omega \cup \alpha = \Omega$. Thus, $\mathcal{F}_A(0) \cap \beta \subseteq \mathcal{F}_A(x) \cup \alpha$. Let for any $x, y, z \in A$ and if $(x * y) * z \in A$, $z \in A$, then $(x * (y * (y * x))) \in A$, and thus, $\mathcal{F}_A((x * y) * z) = \mathcal{F}_A(z) = \mathcal{F}_A(x * (y * (y * x))) = \Omega$. Then, $\mathcal{F}_A(x * (y * (y * x))) \cap \beta = \Omega \cap \beta = \Omega$ and $\mathcal{F}_A((x * y) * z) \cup \mathcal{F}_A(z) \cup \alpha = \Omega \cup \alpha = \Omega$, which indicates that $\mathcal{F}_A(x * (y * (y * x))) \cap \beta \subseteq \mathcal{F}_A(x * y) * z) \cup \mathcal{F}_A(z) \cup \alpha$. Now, if $x \notin A$, then $0 \in A$ or $0 \notin A$, and so, $\mathcal{F}_A(0) \cap \beta = \Omega \cap \beta = \Omega$ or $\mathcal{F}_A(0) \cap \beta = \Gamma \cap \beta = \Gamma$, but $\mathcal{F}_A(x) \cup \alpha = \Gamma \cup \alpha = \Gamma$, which implies that $\mathcal{F}_A(0) \cap \beta \subseteq \mathcal{F}_A(x) \cup \alpha$. Now, if $(x * y) * z \notin A$ or $z \notin A$, then $(x * (y * (y * x))) \in A$ or $(x * (y * (y * x))) \notin A$, and so, $\mathcal{F}_A(x * (y * (y * x))) \cap \beta = \Omega \cap \beta = \Omega$ or $\mathcal{F}_A(x * (y * (y * x))) \cap \beta = \Gamma \cap \beta = \Gamma$, but $\mathcal{F}_A((x * y) * z) \cup \mathcal{F}_A(z) \cup \alpha = \Gamma \cup \alpha = \Gamma$, which implies that $\mathcal{F}_A(x * (y * (y * x))) \cap \beta \subseteq \mathcal{F}_A((x * y) * z) \cup \mathcal{F}_A(z) \cup \alpha$. Hence, $\mathcal{F}_A$ is an $(\alpha, \beta)$-US commutative ideal of $A$ over $U$.

Conversely, assume that $\mathcal{F}_A$ is an $(\alpha, \beta)$-US commutative ideal of $A$ over $U$. If $x \in A$, then $\mathcal{F}_A(0) \cap \beta \subseteq \mathcal{F}_A(x) \cup \alpha = \Omega \cup \alpha = \Omega$. However, $\alpha \subseteq \Omega \subseteq \Gamma \subseteq \beta$; hence, $\mathcal{F}_A(0) = \Omega$, and so, $0 \in A$. Again, if $(x * y) * z \in A$ and $z \in A$, then $\mathcal{F}_A(x * (y * (y * x))) \cap \beta \subseteq \mathcal{F}_A((x * y) * z) \cup \mathcal{F}_A(z) \cup \alpha = \Omega \cup \alpha = \Omega$, and thus, $\mathcal{F}_A(x * (y * (y * x))) = \Omega$, which implies that $(x * (y * (y * x))) \in A$. Therefore, $A$ is a commutative ideal of $E$. $\square$

**Theorem 20** (Extension property). *Let E be a BCK-algebra. For two given subalgebras A and B of E, let $\mathcal{F}_A, \mathcal{F}_B \in S(U)$ such that*
*(i) $\mathcal{F}_A \subseteq \mathcal{F}_B$,*
*(ii) $\mathcal{F}_B$ is an $(\alpha, \beta)$-US ideal over U.*
*If $\mathcal{F}_A$ is an $(\alpha, \beta)$-US commutative ideal over U, then $\mathcal{F}_B$ is also an $(\alpha, \beta)$-US commutative ideal over U.*

**Proof.** Let $\delta \in U$ be such that $B(\mathcal{F}_A : \delta) \neq \emptyset$. By Condition $(ii)$ and Theorem 10, we see that $B(\mathcal{F}_A : \delta)$ is an ideal. We now consider $\mathcal{F}_A$ to be an $(\alpha, \beta)$-US commutative ideal of $A$ over $U$, then $B(\mathcal{F}_A : \delta)$ is a commutative ideal of $A$. Let $x, y \in A$ and $\delta \subseteq \beta$ be such that $x * y \in B(\mathcal{F}_A : \delta)$. Since $(x * (x * y)) * y = (x * y) * (x * y) = 0 \in B(\mathcal{F}_A : \delta)$, it follows from $(C_8)$ and $(i)$ that $(x * (y * (y * (x * (x * y))))) * (x * y) = (x * (x * y)) * (y * (y * (x * (x * y)))) \in B(\mathcal{F}_A : \delta) \subseteq B(\mathcal{F}_B : \delta)$. We see that:

$$x * (y * (y * (x * (x * y)))) \in B(\mathcal{F}_B : \delta) \tag{1}$$

as $B(\mathcal{F}_B : \delta)$ is an ideal and $x * y \in B(\mathcal{F}_B : \delta)$. Furthermore, it is noted that $x * (x * y) \leq x$, and so, we have $y * (y * (x * (x * y))) \leq y * (y * x)$ by $(C_7)$. Thus,

$$x * (y * (y * x)) \leq x * (y * (y * (x * (x * y)))) \tag{2}$$

Hence, by using (1) and (2), we get $x * (y * (y * x)) \in B(\mathcal{F}_B : \delta)$. Therefore, $B(\mathcal{F}_B : \delta)$ is a commutative ideal, and so, $\mathcal{F}_B$ is an $(\alpha, \beta)$-US commutative ideal over $U$ by Theorem 17. $\square$

## 7. Conclusions

Soft set theory is an important mathematical notion, which easily handles uncertainties and has applications in real-life problems. In this paper, we introduce the notions of $(\alpha, \beta)$-US sets in $BCK/BCI$-algebras and $(\alpha, \beta)$-US ideals and $(\alpha, \beta)$-US commutative ideals of $BCK$-algebras. We also

investigate some of their characterizations in detail. We hope that the results given in this paper will have an impact on the upcoming research in this area and other aspects of soft algebraic structures so that this leads to new horizons of interest and innovations. Our results can also be applied to other algebraic structures, such as an $(\alpha, \beta)$-US hemiring, an $(\alpha, \beta)$-US topology, $(\alpha, \beta)$-US $B$-algebras, $(\alpha, \beta)$-US $KUS$-algebras, $(\alpha, \beta)$-US Vector algebras, and $(\alpha, \beta)$-US lattices, and can be applied to other branches of pure mathematics. In addition, the recent development of fuzzy soft-covering-based [19,21,51] problems and $\beta$-covering-based rough fuzzy covering [23,52,53] problems have a huge scope of application to develop fuzzy soft covering $BCK/BCI$-algebras, fuzzy soft $\beta$-covering-based rough fuzzy $BCK/BCI$-algebras, as well as the construction of other fuzzy covering algebras.

**Author Contributions:** Conceptualization, C.J.; formal analysis, M.P.; writing, original draft preparation, C.J.; writing, review and editing, C.J. and M.P.; supervision, M.P.

**Funding:** This research received no external funding.

**Conflicts of Interest:** The authors declare no conflict of interest.

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
