# Peer review of "On (α,β)-US Sets in BCK/BCI-Algebras"

_mathematics, doi:10.3390/math7030252_

Round 1

Reviewer 1 Report

I reommend it for publication.

Author Response

Reviewer 1:

Comments and Suggestions for Authors:

I reommend it for publication.

Answer: Thank you, Sir, for your recommendation for ”publication”.

Reviewer 2 Report

This paper has two problems:
(1) It has both too many grammatical typos and small numbers of mathematical typos. Especially, section 5 should be consulted with a native English speaker, and in the proof of Theorem 6, they mentioned "Theorem 9" which shows the organization seems to be absurd.
(2) It needs to be shorten the content of the manuscript, i.e., the authors re-organize their paper in order to include only the core of the contents after deleting not-so-much important parts of the manuscript.

On the basis of this reason, I recommend this paper to be resubmitted after nishing the above points.

Author Response

Reviewer 2:

Comments and Suggestions for Authors:

This paper has two problems:

(1) It has both too many grammatical typos and small numbers of mathematical typos. Especially, section 5 should be consulted with a native English speaker, and in the proof of Theorem 6, they mentioned Theorem 9 which shows the organization seems to be absurd.

(2) It needs to be shorten the content of the manuscript, i.e., the authors re-organize their paper in order to include only the core of the contents after deleting not-so-much important parts of the manuscript.

On the basis of this reason, I recommend this paper to be resubmitted after nishing the above points.

Reply: Thank you sir, for your valuable comments and helpful for the improvement of the proposed paper.

(1) As per your suggestion, I remove typos and grammatical errors, and corrected the Theorem 6.

(2) For your recommendation, I minimize the context of the manuscripts with the deletion of Definition 4,5; Proposition 5; Theorem 9 for which does not violet main contribution of the manuscript.

Reviewer 3 Report

In this paper authors introduce the notion of (α, β)-Uion-Soft ((α, β)-US) set and with supporting some examples. Then, they discuss the soft BCK/BC I-algebras which are called (α, β)-US algebras, (α, β)-US ideals, (α, β)-US closed ideals and (α, β)-US commutative ideals. In particular, some related properties and relationships of the algebraic structures are investigated. They also provide the condition of an (α, β)-US ideal to be an (α, β)-US closed ideal. Some conditions for a Union-Soft (US) ideal to be a US commutative ideal are given by means of (α, β)-Unions. Moreover, several characterization theorems of (closed) US ideal and US commutative ideals are given in terms of (α, β)-Unions. Finally, the extension property for an (α, β)-US commutative ideal is established.

The paper is, in my idea, interesting and correct.

I recommend publication.

Author Response

Reviewer 3:

Comments and Suggestions for Authors:

In this paper authors introduce the notion of (α, β)-Uion-Soft ((α, β)-US) set and with supporting some examples. Then, they discuss the soft BCK/BC I-algebras which are called (α, β)-US algebras, (α, β)-US ideals, (α, β)-US closed ideals and (α, β)-US commutative ideals. In particular, some related properties and relationships of the algebraic structures are investigated. They also provide the condition of an (α, β)-US ideal to be an (α, β)-US closed ideal. Some conditions for a Union-Soft (US) ideal to be a US commutative ideal are given by means of (α, β)-Unions. Moreover, several characterization theorems of (closed) US ideal and US commutative ideals are given in terms of (α, β)-Unions. Finally, the extension property for an (α, β)-US commutative ideal is established.

The paper is, in my idea, interesting and correct.

I recommend publication.

Answer: We are thankful to the respected reviewer for your comments ”interesting and correct”, and recommendation for ”publication”. Sir, as per your suggestion, we try to in our best to improve the paper.

Reviewer 4 Report

The authors of the paper describe their proposed approach for On (\alpha,\beta)-Union-soft sets in BCK/BCI-algebras . The topic is interesting and with possible applicability. However, the paper needs several improvements:

1) the main contribution and originality should be explained in more detail

2) the motivation of the approach  needs further clarification for benefit of the readers

3) discussion of related work should be expanded with more recent work

4) Minor grammar and syntax issues need correction

5) more examples to illustrate the ideas presented in the paper are needed

6) the conclusions should be extended with more future work

7) More references to recent related papers should be included

8) In general, the paper should be more friendly to the users

Author Response

Reviewer 4:

Comments and Suggestions for Authors:

The authors of the paper describe their proposed approach for On (α, β)-Union-soft sets in BCK/BCI-algebras. The topic is interesting and with possible applicability. However, the paper needs several improvements:

1) the main contribution and originality should be explained in more detail

2) the motivation of the approach needs further clarification for benefit of the readers

3) discussion of related work should be expanded with more recent work

4) Minor grammar and syntax issues need correction

5) more examples to illustrate the ideas presented in the paper are needed

6) the conclusions should be extended with more future work

7) More references to recent related papers should be included

8) In general, the paper should be more friendly to the users.

Reply 1: We rewrite the main contribution and originality of the paper.

Reply 2: We rewrite the motivation part of the paper again.

Reply 3: As per your suggestion, we discuss the paper with recent some papers.

Reply 4: We remove the minor grammatical mistake.

Reply 5: We introduced a new Example 8 for (α, β)-US ideal is closed.

Reply 6: We rewrite the conclusion portion with some future recommendation works.

Reply 7: We add some recent published related to Soft-union algebras and (α, β)-intersectional soft algebra’s papers to the reference.

Reply 8: We are thankful to the respected reviewer his comments ”topic interesting with possible applicability”. Sir, as per your suggestion, we try to in our best to improve the paper.

We tried our best to improve the manuscript and made some Corrections based on reviewers’ comments and suggestions. We would like to appreciate for Editors/Reviewers’ warm work earnestly, and hope that the correction will meet with approval. Once again, thank you very much for your comments and suggestions.

Best Regards

Chiranjibe Jana

Round 2

Reviewer 2 Report

I recommend this paper to be published in the journal.

Reviewer 4 Report

The authors have made the suggested changes and the paper can be accepted

This manuscript is a resubmission of an earlier submission. The following is a list of the peer review reports and author responses from that submission.

Round 1

Reviewer 1 Report

1. The research motive is not clear, please explain the importance.

2. The presentation is weak, please improve it.

3. Please add some decision-making applications to show your motive. Maybe the following references are useful:

1. H. Jiang, J. Zhan, D. Chen, Covering based variable precision (I,T)-fuzzy rough sets with applications to multi-attribute decision-making, IEEE Transactions on Fuzzy Systems, DOI 10.1109/TFUZZ.2018.2883023.

2. X. Ma, J. Zhan, M.I. Ali, N. Mehmood, A survey of decision making methods based on two classes of hybrid soft set models, Artificial Intelligence Review, 49(4)(2018), 511-529.

3. J. Zhan, B. Sun, J.C.R. Alcantud, Covering based multigranulation (I,T)-fuzzy rough set models and applications in multi-attribute group decision-making, Information Sciences, 476 (2019) 290–318.

4. L. Zhang, J. Zhan, Z.X. Xu, Covering-based generalized IF rough sets with applications to multi-attribute decision-making, Information Sciences, 478 (2019) 275-302.

5. J.  Zhan, W.  Xu, Two types of coverings based multigranulation rough fuzzy sets and applications to decision making, Artificial Intelligence Review, 2018,

https://doi.org/10.1007/s10462-018-9649-8

6. L. Zhang, J. Zhan, Novel classes of fuzzy soft β-coverings-based fuzzy rough sets with applications to multi-criteria fuzzy group decision making, Soft Computing, 2018, https://doi.org/10.1007/s00500-018-3470-9.

7. L. Zhang, J. Zhan, Fuzzy soft β-covering based fuzzy rough sets and corresponding decision-making applications, Int. J. Mach. Learn. Cybern., 2018,doi/10.1007/s13042-018-0828-3.

8. J. Zhan, Q. Wang, Certain types of soft coverings based rough sets with applications, Int. J. Mach. Learn. Cybern., 2018,doi/10.1007/s13042-018-0785-x.